# Isolated Severe Dysphonia as a Presentation of Post-COVID-19 Syndrome

**DOI:** 10.3390/diagnostics12081839

**Published:** 2022-07-29

**Authors:** Joanna Jeleniewska, Ewa Niebudek-Bogusz, Jakub Malinowski, Joanna Morawska, Joanna Miłkowska-Dymanowska, Wioletta Pietruszewska

**Affiliations:** 1Department of Otolaryngology, Head and Neck Oncology, Medical University of Lodz, 90-419 Lodz, Poland; jz.jeleniewska@gmail.com (J.J.); ewa.niebudek-bogusz@umed.lodz.pl (E.N.-B.); jtkmalinowski@gmail.com (J.M.); wioletta.pietruszewska@umed.lodz.pl (W.P.); 2Department of Pneumology and Allergy, Medical University of Lodz, 90-419 Lodz, Poland; joanna.milkowska-dymanowska@umed.lodz.pl

**Keywords:** high-speed videolaryngoscopy, HSV, dysphonia, post-COVID-19, larynx, self-assessment of voice

## Abstract

This is the first study assessing the clinical management of severe, isolated dysphonia during post-COVID-19 syndrome. One hundred and fifty-eight subjects met the inclusion criteria for the post-COVID-19 condition as specified by the WHO. Six patients were diagnosed with isolated severe dysphonia, constituting 3.8% of the initial group. The pre- and post-examination protocol consisted of subjective voice self-assessment and routine laryngological examination, followed by an instrumental examination by means of Laryngovideostroboscopy (LVS) and High-Speed Videolaryngoscopy (HSV). The treatment included short-term systemic steroids in decreasing doses, moisturizing inhalations with hyaluronic acid, and protective agents against Laryngopharyngeal Reflux. The kinematic imaging of the glottis performed by means of HSV before treatment showed deviations in the regularity and symmetry of vocal fold vibrations, absence of mucosal wave, and incomplete glottal closure. Improvement of the structural and functional state of the larynx was observed post-treatment. Kymographic sections and Glottal Width Waveform (GWW) graphs obtained from post-treatment HSV recordings showed improvement in vocal fold vibrations. The decrease in mean Jitter and Shimmer was observed, with the following mean values of 3.16 pre-treatment and 2.97 post-treatment for Jitter and 7.16 pre-treatment and 2.77 post-treatment for Shimmer. The post-treatment self-evaluation of voice showed considerable improvement in vocal function and voice quality in all the examined patients. Severe dysphonia in patients with post-COVID-19 syndrome requires urgent ENT diagnosis using instrumental assessment with the evaluation of laryngeal phonatory function and intensive comprehensive treatment.

## 1. Introduction

COVID-19 is a severe disease caused by the SARS-CoV-2 virus, which since its first reported cases from Wuhan, Hubei Province, China at the end of 2019 has spread rapidly worldwide [1,2,3]. The disease may be asymptomatic or may range from mild to very severe symptoms, including multiorgan dysfunction or fatal pneumonia [4,5].

The virus is transmitted between humans via droplets generated during coughing or sneezing and direct contact with mucous membranes [6]. The most common symptoms are fever, cough, shortness of breath, muscle ache, and less often nausea, vomiting, or diarrhea [3].

Patients affected by COVID-19 may also suffer from the nonspecific symptoms of upper airway infection, such as rhinorrhea, nasal obstruction, pharyngitis, and laryngitis. A single study analyzed the occurrence of otolaryngological symptoms in 57.4% of patients with the results as follows: the most frequent anosmia (35.4%), sore throat (27%), ageusia (16.1%), nasal congestion (12.9%), postnasal discharge (6%), otalgia (2%), runny nose (9%), tinnitus (1.2%), hoarseness (5.1%), gingivitis (1.2%), Bell’s palsy (0.6%), sudden hearing loss problems (0.6%) [2]. Some patients report mainly complaints related to laryngeal involvement in the inflammatory process during the COVID-19 disease such as hoarseness, feeling of dryness in the larynx, voice fatigue, or aphonia. These symptoms, and characteristic features of dysphonia were described among COVID-19 patients with different prevalence rates (22–26%) among various nations [1,7]. Another study regarding vocal symptoms analyzed the prevalence of voice disorders in 702 patients with mild to moderate COVID-19 from 19 European hospitals and showed that 27% reported dysphonia during the clinical course of the disease [1].

In some patients, symptoms last for far longer than expected. Throughout the pandemic, various terminology was used to describe illness in patients who have recovered from COVID-19 but are still reporting lasting effects of the infection such as long COVID, long-haul COVID, or recommended by WHO, post-COVID-19 condition. A clinical case definition of post-COVID-19 condition achieved by a Delphi consensus was published by WHO and reads as follows: “post-COVID-19 condition occurs in individuals with a history of probable or confirmed SARS-CoV-2 infection, usually 3 months from the onset of COVID-19 with symptoms that last for at least 2 months and cannot be explained by an alternative diagnosis.” The authors who reached the Delphi Consensus indicated dysphonia as one of the conditions associated with post-COVID-19 syndrome [8].

Other authors underlined that dysphonia was a long-lasting symptom found in patients affected by mild-to-moderate COVID-19. The studies concerning voice disorders in the course of COVID-19 disease were mainly based on subjective methods of examination [8,9,10]. The use of instrumental methods of assessment in diagnosing dysphonia in COVID-19 patients, namely, videolaryngoscopy, fluoroscopy, fibro-optic laryngoscopy, was reported only by a few researchers [11,12,13,14]. Instrumental data on dysphonia after COVID-19 are not plentiful and only emerging now because of restrictions in the use of endoscopy at the outset of the pandemic [14]. Moreover, the methods applied in these studies did not allow for functional assessment of the phonatory function in a way that High-Speed Videolaryngoscopy (HSV) does. In the currently available literature, a study including dynamic evaluation of vocal fold phonatory movements in dysphonic COVID-19 patients has not been found.

The aim of the present study was to assess the incidence and clinical management of isolated severe dysphonia during post-COVID-19 syndrome.

## 2. Materials and Methods

### 2.1. Patients

Included in the study were 158 patients: 83 men (mean age: 48.2 years; range 35–65; SD 1.09) and 75 women (mean age: 55.6 years; range 38–66; SD 0.98). All patients attended an ENT outpatient clinic and/or post-COVID-19 outpatient clinic in Norbert Barlicki Memorial Teaching Hospital No. 1 of the Medical University of Lodz, Poland, because of persistent symptoms.

The inclusion criteria were as follows: both sexes, age > 18 years, a diagnosis of COVID-19 infection based on WHO guidelines confirmed by reverse transcription polymerase chain reaction (RT-PCR) analysis, fitting the criteria of post-COVID-19 condition as specified by the WHO [8], and signed informed consent.

Exclusion criteria were history of laryngeal disorders before SARS-CoV19 infection, severe course of the disease, invasive mechanical ventilation, history of diseases that may have impacted voice quality before COVID-19 such as Laryngopharyngeal Reflux (LPR), acute sinusitis, exacerbation of chronic sinusitis due to reasons other than COVID-19, chronic laryngeal disorders or surgical laryngeal procedures. Occupational voice users were excluded to rule out previous work-related injuries to the voice organ.

First, from the group of 158 patients, those with otolaryngological symptoms were selected (N = 48; 30.38%). The symptoms most frequently listed by these patients were: sudden deafness, tinnitus, vertigo, prolonged loss of smell and taste, postnasal drip, and/or nasal blockage as the symptoms of sinusitis. Within that group, 28 patients (17.71%) had dysphonia and other ENT symptoms mostly related to Laryngopharyngeal Reflux disease; that is, heartburn and regurgitation. Only six patients (3.8%) presented with dysphonia as an isolated symptom and therefore met the inclusion criteria listed above (Figure 1).

Informed consent was sought, and confidentiality was ensured by a numerical cross-referencing system. Data collection took place from January 2021 till the end of January 2022. Approval for this study was granted by the Ethical Committee of the Medical University of Lodz (decision no RNN-96/20KE).

### 2.2. Methods

Medical history including co-morbidity of other systemic and/or chronic laryngological disorders was gathered, and sociodemographic data were collected from all the participants of the study, including information on sex, age, smoking history, the course of the disease, and the applied treatment.

The examination procedure included subjective voice self-assessment and routine laryngological examination, followed by instrumental examination: Laryngovideostroboscopy (LVS) and High-Speed Videolaryngoscopy (HSV). The examination was performed in the Department of Otolaryngology, Head and Neck Oncology, Medical University of Lodz, Poland.

### 2.3. Subjective Voice Assessment

The subjects completed two voice assessment questionnaires: Voice Handicap Index (VHI) [15] and Voice-Related Quality of Life (V-RQOL) [16]. The VHI is a self-rating test evaluating the biopsychosocial impact of voice problems. It consists of 30 questions divided into 3 domains: physical (VHI-P), emotional (VHI-E), and functional (VHI-F). Each question is scored on a 5-point Likert scale and a higher number of points indicates a more severe voice disorder. The maximum possible score is 120 points. A VHI total score below 30 is considered a low score, meaning that the handicap associated with voice disorder is minimal. Moderate voice handicap is indicated by the total score between 31 and 60 and a VHI total score from 61 to 120 is considered a severe voice handicap. Voice-Related Quality of Life (V-RQOL) is a shorter questionnaire, it consists of 10 questions in total, and it focuses more on the quality of life than the handicap itself. Each statement is scored on a 5-point scale, and then a total score, ranging from 10 to 50, is calculated using the algorithm suggested by the authors of the questionnaire [16]. A higher score indicates a better voice-related quality of life. The participants answered both questionnaires during one evaluation session, prior to the laryngological examination. Assistance with completing the questionnaires was provided when necessary. The same set of questionnaires was used for post-therapy self-assessment of voice.

### 2.4. Instrumental Voice Assessment

First, laryngeal endoscopy was performed using the rigid Olympus HD Laryngoscope 90 endoscope (WA96105A) with a CV-260SL processor and CLV-260SL light source, from Olympus Optical Co. Ltd., Tokyo, Japan. For digital recording of the kinematic image of the larynx we used an Olympus Visera Elite OTV-S190 camera with a xenon lamp providing white light and an Olympus CLL-S1 strobe lamp with a stroboscopic flash rate of 1.5 Hz below the fundamental vibration frequency of the vocal folds. Vocal fold function was assessed using stroboscopic light during repeated stable phonation of a sustained vowel /i:/.

The HSV images were recorded using the Advanced Larynx Imager System and a High-Speed camera (ALIS Cam HS-1, Diagnova Technologies, Wroclaw, Poland). Standard parameters of the vocal fold vibrations were assessed: regularity and symmetry of vibrations, mucosal wave, and the shape of glottal closure. Based on HSV recordings, the kymographic section plots were generated and graphic analysis of vocal folds movement was performed creating a graph called a phonovibrogram. Phonovibrograms are an efficient way to condense and to graphically present in one color map the information about VF function at a given time interval. In the graph, the *X* axis represents time, and the *Y* axis represents points along the edge of vocal folds. The upper part of the graph relates to the left and the lower part to the right vocal fold. The intensity of the red color in the phonovibrogram map denotes the degree of opening of the right or the left fold (the momentary distance between the VF edge and the glottis axis). A black color means that the distance equals zero—the vocal fold is aligned to the center of the glottis. This graph contains a clean visual presentation of most of the data needed to diagnose the patient. Then, Glottal Width Waveform (GWW) graphs were created, which reflected instantaneous changes in the glottal width at different time points. GWW graphs were used to determine parameters describing the disturbances of vocal fold vibrations including period and amplitude perturbation measures: Jitter, PPF (Period Perturbation Factor), PPQ (Period Perturbation Quotient), Shimmer, APQ (Amplitude Perturbation Quotient), APF (Amplitude Perturbation Factor). The parameters were used to measure disturbances in the middle part of the glottis during a glottal cycle. All the above-mentioned parameters are described in detail in the literature on the subject [17,18,19].

### 2.5. Treatment

The treatment of dysphonia in the described COVID-19 patients included short-term systemic steroids—Methylprednisolone p.o. starting at 32 mg per day in descending doses for two weeks, moisturizing inhalations with hyaluronic acid, and protective agents (with hyaluronic acid, sulphate, and Poloxamer) against Laryngopharyngeal Reflux commonly co-occurring in infections of the upper respiratory tract. The pharmacological therapy was complemented by education on vocal hygiene and coping strategies for dealing with stress caused by the incidence of COVID-19 and with long-term compromised vocal capacity. In all subjects, the laryngological examination, self-assessment of voice, and instrumental examination were repeated two weeks and one month post-treatment. After 2 weeks of treatment, the patients were examined to determine whether the applied treatment was having an effect, and a detailed evaluation with the determination of parameters was carried out after 1 month.

## 3. Results

### 3.1. Patients’ Symptoms

All patients (N = 6) with isolated dysphonia had severe voice symptoms confirmed with subjective and objective methods of examination. The pre-treatment examination history showed that the patients suffered from complete voice loss (aphonia), or severe persistent hoarseness accompanied by vocal fatigue and loss of voice strain and stamina. Other commonly reported symptoms included dryness of the throat and larynx, frequent throat clearing, and cough. Following pharmacological treatment, marked alleviation of clinical symptoms and improvement in the morphological condition of the glottis were observed, resulting in improved phonatory function.

### 3.2. Subjective Voice Assessment

The post-treatment self-evaluation of voice showed considerable improvement of the vocal function and voice quality in all the examined patients. It was reflected in the results of VHI and V-RQOL questionnaires, with mean VHI values of 62 and 34 before and after treatment, respectively, and similarly mean V-RQOL values of 57 and 78.

In the case of the VHI questionnaire, noticeable improvement was observed in both the total score and all domains of the test. All patients in this study reported moderate or severe voice disorders, as there were no total VHI scores below 30 points in the pre-treatment self-evaluation of voice. The highest value of the total VHI score was 76 points. Post-treatment results showed that there were no total scores > 60 points reported by any of the patients. Improvement was observed in all patients, and to the most degree in Patient 1, with a difference of 36 points in total VHI pre- and post-treatment scores.

Similarly, the improvement was noticed for the total V-RQOL score—higher post-treatment results indicating an increased voice-related quality of life were observed in all patients, with the highest post-treatment voice-related quality of life reported by Patient 3. Detailed results of the pre- and post-treatment self-assessment of voice are presented in Figure 2.

### 3.3. ENT Examination with Instrumental Assessment including VLS and HSV

The most common findings in ENT examinations were signs of subacute pharyngolaryngitis with evidence of chronic erythema and dryness of the mucosa. The vessels of vocal folds were dilated. The vocal folds appeared reddened, dry, and stiff. In some patients, mucosal and glandular atrophy was associated with thick “frothy” mucus on the vocal fold margins (Figure 2). The described stiffness and mass abnormalities of the vocal folds resulted in disturbances of the phonatory function. In the studied patients, due to the high degree of dysphonia preventing stable phonation, it was not possible to obtain reliable stroboscopic images. High speed video technology overcomes the limitations of videostroboscopy. The kinematic imaging of the glottis during phonation performed by means of HSV in pre-treatment examination showed deviations in the regularity and symmetry of vocal fold vibrations, absence of mucosal wave, and incomplete glottal closure in all the examined patients.

Figure 3 presents the results of HSV pre-treatment examination subjected to kymographic analysis of Subject 2. The kymographic sections obtained in the posterior, middle, and anterior one-third of the length of the vocal folds showed an incomplete glottal closure in the intermembranaceus part of glottis. Other disturbances of phonatory function visible in kymographic analysis included periodic irregularity of the vocal fold vibrations, particularly in the middle part of the glottis (Figure 3B) (marked with an ellipse in the image) and reduction in vibratory amplitude and mucosal wave.

Figure 3D–F shows GWW (Glottal Width Waveform) graphs obtained from a videokymogram at the cross-sections in the posterior, middle, and anterior part of the glottis reflecting changes in the glottal width during recorded glottal cycles. The GWW graphs indicate irregularly changing glottal width within a glottal cycle during steady phonation.

The phonovibrogram (Figure 4A) supports this observation and represents the greatest disturbances in the phonatory oscillation of both vocal folds, particularly in the middle part of the glottis—black notches in the middle of both folds present delayed and slower movement in this area. The stiffness of the vocal folds due to inflammation and thick mucus lying mainly in this area of glottis can cause these disorders. The observed abnormal findings were confirmed by quantitative parameters measuring vocal fold vibrations (see Table 1: Subject 2—the pre-therapy column).

In other patients, the majority of parameters determining the regularity and amplitude of vocal fold vibrations also exceeded thresholds of the norm, particularly Shimmer—the amplitude perturbation parameter describing mean normalized dynamic range between cycles. Similarly, all the other parameters of the Shimmer groups assumed high values in almost all the examined subjects (Table 1—pre-therapy column). High values of amplitude perturbation measures may be associated with changes in the stability of the voice.

After the applied treatment, the improvement of the structural and functional state of the larynx was observed (Figure 5). The symptoms of glottis inflammation decreased or disappeared. The hyperemia of vocal folds and dryness of the mucous membrane diminished. A less thick consistency of the mucus was observed than before treatment, which was important in protecting the glottis from phonotrauma. The improvement of the morphological condition of the glottis contributed to a better phonatory function.

The kymographic sections and GWW graphs obtained from post-treatment HSV recordings show improvement of vocal fold vibrations. The post-treatment phonovibrogram (Figure 4B) indicates better phonatory movements of both vocal folds during recorded glottal cycles.

The results presented in Table 1 support these observations. Table 1 shows differences in values of parameters quantifying vocal fold vibrations in dysphonic COVID-19 patients in the pre- and post-treatment evaluation. The values of amplitude perturbation measures, Shimmer and APQ, improved in all the examined subjects. This observation may reflect better phonatory efficiency of the glottis and better stability of the voice after the inflammation of the glottis had subsided.

## 4. Discussion

One of the otorhinolaryngological complications of COVID-19 is dysphonia, which has been mostly described in the subject literature as a disease associated with mechanical ventilation and cough in the course of COVID-19 infection affecting the lower respiratory system, mainly in hospitalized patients [2,14,20,21]. The presented six non-hospitalized subjects suffered from persistent severe voice disorders that occurred during a mild-to-moderate course of COVID-19. The authors of multicenter research, Lechien et al., 2021 [22], indicate that 3.7% of patients reported severe dysphonia among the studied patients with mild-to-moderate COVID-19. In our studies, the percentage of patients suffering from severe voice disorders in the course of mild-to-moderate COVID-19 is similar—3.80%. The described patients presented with both typical and nonspecific symptoms of upper airway infection, such as rhinorrhea, nasal obstruction, pharyngitis, and laryngitis and were home-managed. Even though many symptoms of COVID-19 infection had subsided, the symptoms of severe dysphonia—aphonia and vocal fatigue—continued for up to 5–6 months and thus the patients fulfilled the diagnostic criteria of post-COVID-19 syndrome. Following comprehensive treatment, significant resolution of clinical symptoms and improvement in the morphological condition of the glottis was observed, resulting in improved phonatory function.

Laryngeal endoscopy with HSV was used for instrumental assessment of the larynx in the preliminary and post-therapy examination. The pre-therapy laryngeal endoscopy showed subacute inflammation of the glottis with evidence of chronic erythema—dryness of the mucosa in all the examined patients. The assessment of vocal fold oscillations by means of Laryngovideostroboscopy (LVS) in the studied subjects failed. It was not possible to obtain reliable stroboscopic images due to the high degree of corditis preventing long, stable, and regular phonation, which is necessary to synchronize the stroboscopic light with the fundamental frequency of the vocal fold vibrations. These problems were overcome with the use of HSV.

Recently, there has been a growing interest in clinical practice in the use of high-speed film to record a real image of the glottis’ phonatory function as this tool allows the recording and analysis of dynamic images of severe vocal fold pathology [17,22,23,24]. This examination can be conducted in cases of irregular, limited vocal fold vibrations, when the patient with severe corditis is unable to maintain a sufficiently long, stable phonation. For comparison, it takes 10 s to record 10 glottal cycles in the LVS, while a high-speed camera (HSV) can record this number of cycles in just 0.1 s {Formatting Citation}. The altered glottal structure and poor/compromised phonatory function in the studied patients were accurately reflected in the HSV images. In the HVS recordings we have found the signs that indicate severe corditis with glottic insufficiency and stiffness of vocal folds. The kymographic analysis shows irregularity of vocal fold vibrations and reduction or loss in vibratory amplitude and mucosal wave. These observations were confirmed by quantitative parameters describing disturbances of vocal fold vibrations including period and amplitude perturbation measures. In the preliminary examination, the values of the parameters describing the disturbances of vocal fold vibrations, particularly Jitter and Shimmer group parameters, considerably exceeded thresholds of the norm. It was particularly marked in the high (exceeding threshold of the norm) values of Shimmer—an amplitude perturbation parameter describing mean normalized dynamic range between glottal cycles. High values of amplitude perturbation measures may be associated with changes in the stability of the voice and loss of voice volume [25,26]. Furthermore, Shimmer measures are associated with the perception of hoarseness of the voice [25]. In the studied patients, the described changes may be reflected by persistent severe dysphonia or voice loss, which are the consequences of laryngological involvement such as long-lasting inflammation. Similarly, Asiaee et al. [26] reported changes in the acoustic parameters of voice in 64 patients diagnosed with COVID-19. The authors revealed higher aperiodicity and irregularity of vocal fold vibrations, reflected in increased levels of Jitter and Shimmer, quantifying the cycle-to-cycle variation in frequency and amplitude, respectively.

It has been reported by a number of authors that severe dysphonia diagnosed in mild-to-moderate COVID-19 patients could be likely related to laryngeal involvement by the airway inflammatory process and may be caused by vocal fold edema or inflammation in the course of COVID-19 infection [1,10,14]. The possible etiology of corditis might be the previous presence of SARS-CoV-2 in the larynx. Studies by Nisreen et al. [27] and Sato et al. [28] have shown the presence of the ACE-2 receptor, considered to be the SARS-CoV-2 receptor, also in the larynx, including the vocal folds. Therefore, dysphonia as a symptom of COVID-19 may occur due to direct entry of SARS-CoV-2 into laryngeal cells, leading to laryngeal and vocal cord inflammation.

ACE-2 is expressed across a wide variety of human tissues, in addition to the lungs, indicating that SARS-CoV-2 may infect other tissues aside from the lungs [29]. The findings of our research are in agreement with the observations of Cantarella et al. (2021) that long-term impairment of the glottis in the course of mild-to-moderate COVID-19 disease is indicative of more severe and longer-lasting disturbance in voice production than is usually seen in viral laryngitis [10]. The examined patients had suffered from severe dysphonia for more than 5 months and they required intensive phoniatric treatment, including pharmacological therapy, education on vocal hygiene, vocal training, and coping strategies for dealing with stress. The last one was an important part of treatment given that the psycho-emotional factors may constitute a considerable risk of persisting voice disorders. After one month of comprehensive treatment, the patients reported symptom relief and the improvement of the phonatory function.

In the post-treatment examination, HSV assessment revealed the improvement of morphology and function of the vocal folds in all the examined subjects. The redness and stiffness of the vocal folds diminished, and the vocal folds were more pliable and their vibratory oscillations became more regular with better amplitude and mucosal wave in comparison to HSV recordings in pre-treatment examination. These observations were confirmed by improvement of the values of the parameters quantifying vocal fold vibrations. The substantial improvement of the value of Shimmer may be associated with better stability of the voice and an increase of voice volume [24,30] observed in patients after applied treatment.

The post-treatment improvement observed in instrumental examination of voice apparatus was reflected in the self-assessment tests: the VHI and V-RQOL questionnaires. When diagnosing or treating patients with voice problems, measurement of the biopsychosocial impact of the voice problem is an indispensable instrument for monitoring the effectiveness of therapy [31]. According to WHO guidelines, the patient’s perspective should always be taken into account in the diagnostic process and self-assessment tools should be an integral part of the thorough and holistic voice diagnosis.

It should be underlined that dysphonia diagnosed in the studied subjects was primary an organic type caused by severe corditis confirmed by means of objective assessment. Interestingly, in a number of studies, the authors reported functional muscle tension dysphonia associated with stress caused by the incidence of COVID-19 with a breathing pattern disorder and cough [14,21,32]. The study patients reported the stress of going through COVID-19 infection and persisting voice disorders; however, the objective assessment of the larynx function ruled out the psychogenic factor as the main risk factor for diagnosed dysphonia.

The study is the first one describing instrumental voice assessment by means of HSV to reveal severe organic dysphonia as a serious disease in post mild-to-moderate COVID-19 patients. Research shows that physicians must keep in mind that COVID-19 patients may develop dysphonia throughout the clinical course of the disease such as in the case of a common viral infection of the upper aerodigestive tract mucosa, but the prognosis is more serious. Therefore, laryngeal imaging using objective instrumental methods such as HSV may be more helpful in distinguishing whether dysphonia is of emotional etiology or results from laryngeal involvement.

## 5. Conclusions

Severe dysphonia in patients with post-COVID-19 syndrome requires urgent ENT diagnosis using instrumental assessment with the evaluation of laryngeal phonatory function and intensive comprehensive treatment. The application of the HSV allows for an accurate and objective assessment of the glottis phonatory function facilitating an in-depth functional diagnostic process, providing essential objective indications for treatment, and enabling monitoring of the therapy effects in patients with post-COVID-19 syndrome.

## Figures and Tables

**Figure 1 diagnostics-12-01839-f001:**
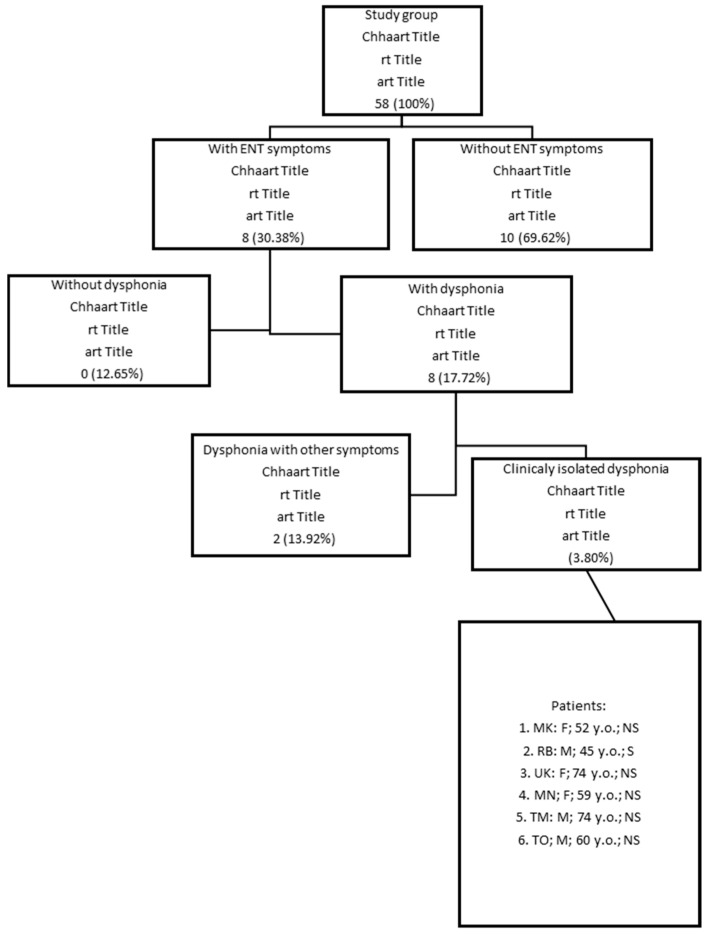
Study group (N = 158): the final rectangle presents the patients with isolated severe dysphonia during post-COVID-19 *syndrome*, and the description includes initials; gender (M—male; F—female); y.o. (years old); smoking status (S—smoker, NS—non-smoker).

**Figure 2 diagnostics-12-01839-f002:**
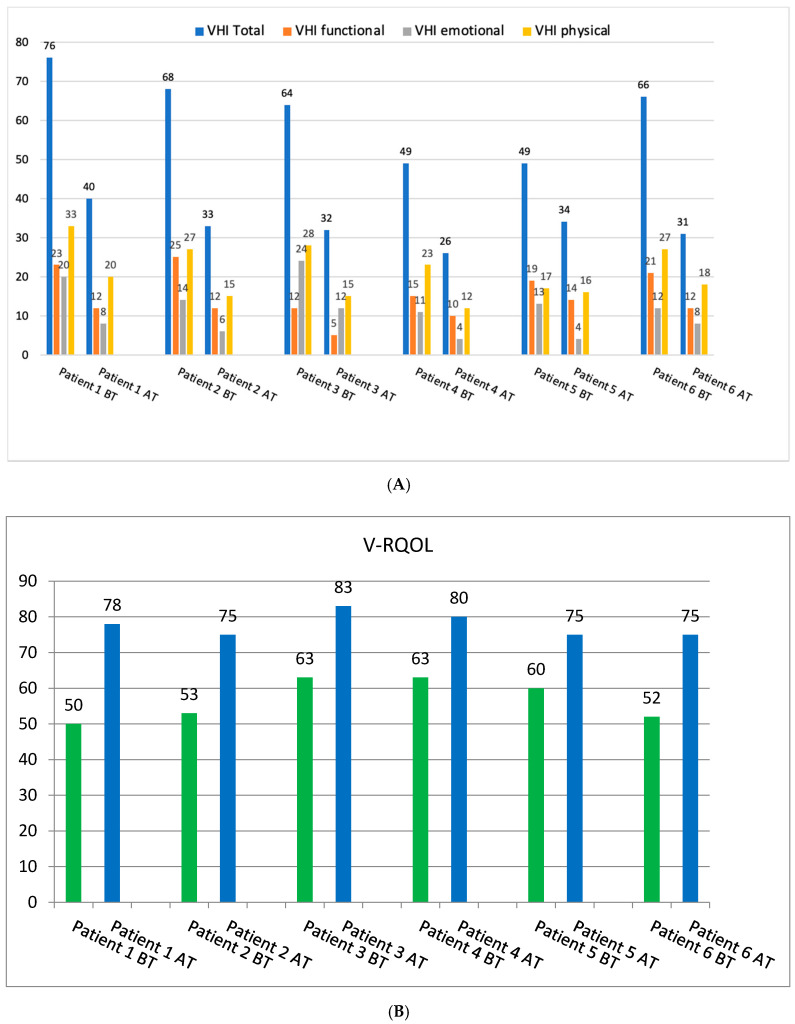
Pre- and post-treatment subjective self-assessment of voice results. (**A**). Pre- and post-treatment scores for the VHI total and domains. (**B**). Pre- and post-treatment scores for the V-RQOL total. BT—before treatment; AT—after treatment.

**Figure 3 diagnostics-12-01839-f003:**
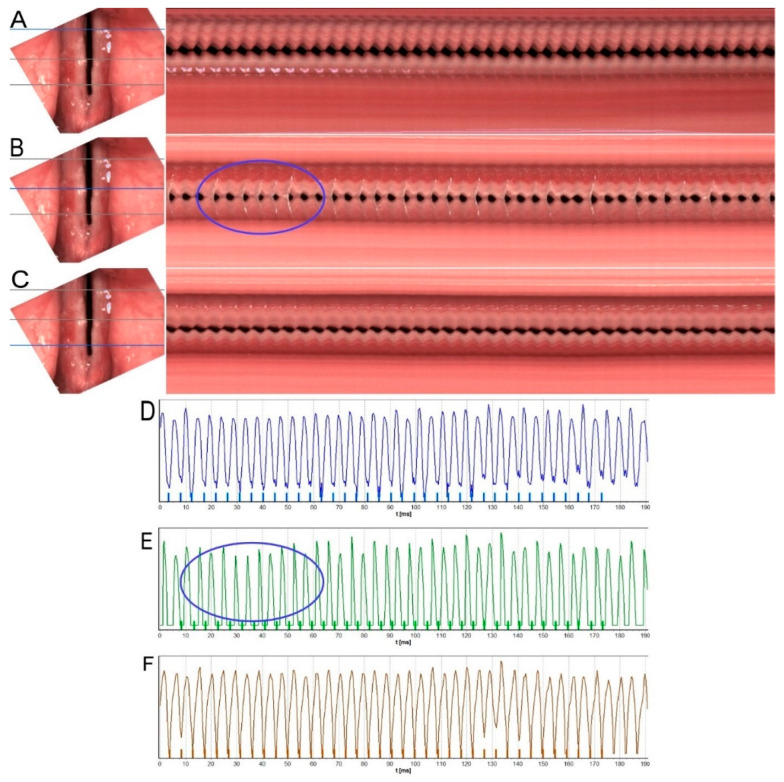
Images of the glottis for Subject 2 before treatment, obtained by HSV recording. (**A**–**C**)—on the left: rotated images of vocal folds with grey lines marking three cross-sections used for generating videokymograms. On the right: long videokymograms obtained for cross-sections marked with a blue line. Respectively: (**A**)—posterior, (**B**)—middle, (**C**)—anterior part of the glottis. Blue ellipse marks most visible irregularity of vocal folds vibration. (**D**–**F**)—Glottal Width Waveform (GWW) plots obtained from corresponding videokymograms: the posterior, middle, and anterior sections of the glottis, respectively. Blue ellipse marks irregularity seen in the middle kymogram.

**Figure 4 diagnostics-12-01839-f004:**
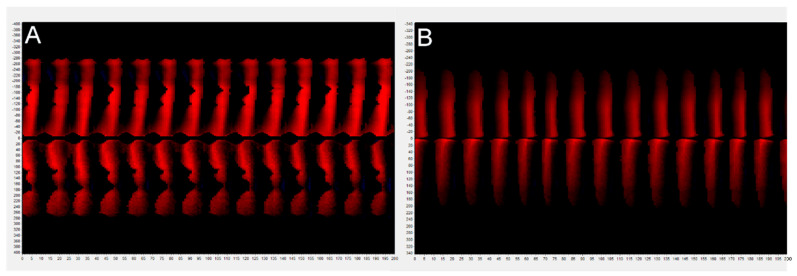
Phonovibrogram generated (**A**)-before treatment, (**B**)-after treatment. Graph presents the phonatory oscillation of vocal folds.

**Figure 5 diagnostics-12-01839-f005:**
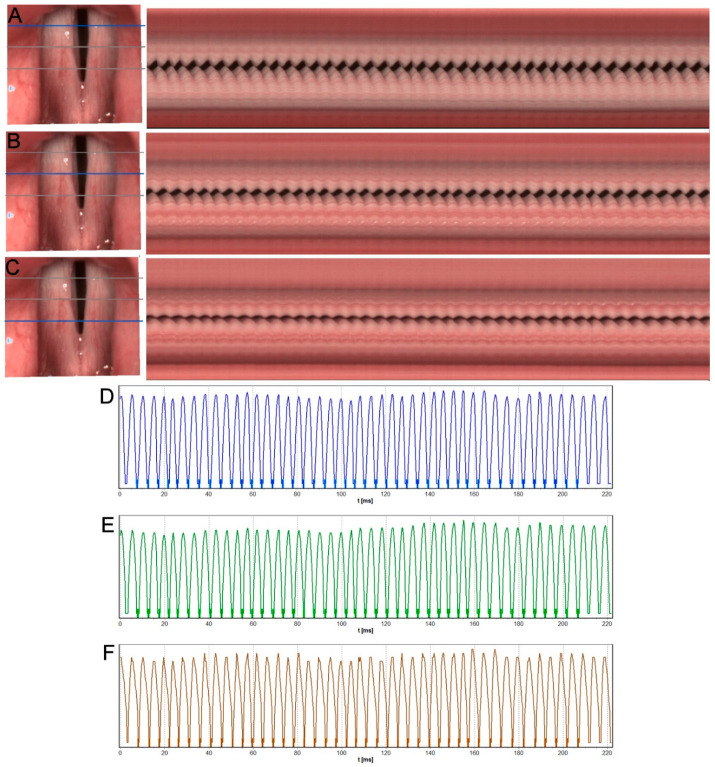
Images of the glottis for Subject 2 after treatment, obtained by HSV recording. (**A**–**C**)—on the left: rotated images of vocal folds with grey lines marking three cross-sections used for generating videokymograms. On the right: long videokymograms obtained for cross-sections marked with a blue line. Respectively: (**A**)—posterior, (**B**)—middle, (**C**)—anterior part of the glottis. (**D**–**F**)—Glottal Width Waveform (GWW) plots obtained from corresponding videokymograms: the posterior, middle, and anterior sections of the glottis, respectively.

**Table 1 diagnostics-12-01839-t001:** Parameters describing abnormalities in frequency and amplitude of phonatory vibrations calculated on the basis of kymograms from HSV recordings.

Patient	BR Subject 1		KU Subject 2		KM Subject 3		MT Subject 4		NM Subject 5		OM Subject 6	
	Pre	Post	Pre	Post	Pre	Post	Pre	Post	Pre	Post	Pre	Post
F0Avg [Hz]	183.5	150	245.7	212	335.1	536.1	427.8	286.8	295.7	215.5	280.9	323.5
PPF [%]	4.13	4.1	1.18	4.86	0.63	5.29	7.56	0.31	3.92	1.76	1.88	2.02
Jitt [%]	4.14	3.96	1.18	4.73	0.63	5.08	7.37	0.31	3.77	1.68	1.88	2.03
APF [%]	4.05	1.96	9.54	3.96	6.58	3.5	26.93	1.64	1.27	3.41	2.92	2.04
Shimmer [%]	4.11	1.94	8.87	4	6.49	3.55	19.29	1.64	1.27	3.44	2.94	2.04
PPQ3 [%]	2.22	2.48	0.5	2.61	0.35	3.08	3.79	0.17	2.19	0.92	1.19	1.25
APQ3 [%]	2.19	1.15	2.4	1.97	4.25	2.04	10.41	0.93	0.5	2.11	1.41	1.17
AmpAvg [%FL]	8.6	8.2	5.7	4.1	6.7	3.8	3.8	4.5	3.8	4	6.1	8.4
AmpAvg 2/3 [%FL]	8.3	11.4	6	3.4	8.5	4.6	3.5	5	4.1	5.5	6.6	9.6
RGGA [%]	0	0	10.7	0	5.9	33.7	0	11.3	0	1.9	0	0
OQAvg [%]	57.8	53.9	73.6	53.2	86.1	99.5	49	89.7	55.7	35.8	65.4	58.5
AmplAsymAvg [%]	13.1	12.2	21.7	6.8	7.3	10.7	19.2	5	9.8	17.4	19.9	6.4
AmplAsym Avg_2/3 [%]	16.3	10.5	14.5	4.8	7.1	6.6	18.5	5.5	7	15.1	19.3	6.3
AbsPhase DiffAvg [°]	34.3	49	115.1	96.2	28.9	24	107	71.8	56.4	53.6	29.3	34.5

F0Avg—Average fundamental frequency; PPF—Period Perturbation Factor; Jitt—Jitter; APF—Amplitude Perturbation Factor; PPQ3—Period Perturbation Quotient; APQ3—Amplitude Perturbation Quotient; RGGA—Relative Glottal Gap Area; OQAvg—Average Open Quotient; AmplAsymAvg—Average Amplitude Symmetry; AbsPhase DiffAvg—Average phase asymmetry.

## Data Availability

Data available on request.

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
