# Peer review of "Isolated Severe Dysphonia as a Presentation of Post-COVID-19 Syndrome"

_diagnostics, 2022, doi:10.3390/diagnostics12081839_

Round 1

Reviewer 1 Report

This is a good article with the only objective data so far on the cause of dysphonia in patients with post-COVID-19 syndrome. Nevertheless, some minor details need to be corrected.

Methods

Lines 120-122: I suggest adding also the explanation of “y.o.” in the Figure legend.

Lines 142-145: »First, laryngeal endoscopy was performed using the rigid Olympus HD Laryngo- 142 scope 90 endoscope (WA96105A). ……..« I suggest adding the producer of the equipment, city and country of the producer of the equipment.

Lines 148-149: »The HSV images were recorded using the Advanced Larynx Imager System and a 148 High-Speed camera (ALIS Cam HS-1, Diagnova Technologies). « I suggest adding the city and country of the producer of the equipment.

Lines 161-162: »The treatment of dysphonia in the described COVID-19 patients included short-term systemic steroids in decreasing doses. « Which corticosteroid was used, what was the way of administration, and what was the dose and length of treatment?

Lines 167-168: » In all subjects, the laryngological examination, self-assessment of voice, and instrumental examination were repeated two weeks and one-month post-treatment. « The authors present only one result for every post-treatment assessment. Was this assessment performed two weeks or one months after the completed treatment? Is the result mean value of two assessments?

Results

Line 171: » 3.1. Patients«   In view of the content I suggest »3.1. Patients' symptoms«

Lines 177-179: »Following pharmacological treatment, significant alleviation of clinical symptoms and improvement in the morphological condition of the glottis were observed, resulting in improved phonatory function. « Did they perform some kind of objectivization of the symptoms and statistical analysis in order to state »significant alleviation«? Otherwise, I would suggest using »marked«, »important« or some other term.

Lines 212-214: »The kinematic imaging of the glottis during phonation performed by means of HSV in pre-treatment examination showed deviations in the regularity and symmetry of vocal fold vibrations, absence of mucosal wave, and incomplete glottal closure. « In how many patients?

Lines 237-239: Table 1. In the Table legend there should be the explanations of all abbreviations used in the table. They should indicate values that exceed the limit of normal values in order to make the Table 1 more understandable. All the presented parameters should be mentioned in the Methods.

Lines 241-243: »The phonovibrogram (Fig 4A) supports this observation and represents the greatest disturbances in the phonatory oscillation of both vocal folds, particularly in the middle part of the glottis. « I suppose that the readers of Diagnostics are not only otorhinolaryngologists. The authors do not mention »phonovibrogram« in the Methods section. The non-expert reader cannot understand it.

Discussion

Lines 357-359: »The decrease in mean Jitter and Shimmer was observed, with the following mean values of 3.16 pre-treatment and 2.97 post-treatment for Jitter and 7.16 pre-treatment and 2.77 post-treatment for Shimmer. «

And Lines 368-371: » All the examined patients reported improved voice conditions and voice-related quality of life in the voice self-assessment questionnaires, with mean VHI values of 62 and 34 before and after treatment, respectively, and similarly mean V-RQOL values of 57 and 78. «  All results should be presented in the Results section. They should not be mentioned for the first time in the Discussion section.

Reviewer 2 Report

the authors present a case series of six patients with pos covid dysphonia. As pointed out by the authors, severe disorders of voice/dysphonia have an impact on quality of life and on communication handicaps. Although not that frequently reported, the high impact makes dysphonia a critical symptom to be searched on. The authors carefully selected the patients, excluding all possible biases such as intubation, reflux disease, and previous voice disorders. The evaluation method included questionnaires of self-perception and related quality of life, as well as acoustic assessment of the voice and dynamic endoscopic evaluation. The authors did not include a perceptual evaluation, such as GRBAS/GRBASI scale, describing the acoustic perception of the voice.

The results were well presented, all data are presented individually allowing a good overview of pre-pos treatment improvement. The figures are well exposed and I consider them essential for the publication.

the table brings all data of the objective analysis.

discussion should include that 2 patients were older than 70 years old, so some alterations of the voice might be related to aging, enhancing the improvement of pre-pos treatment. Also, I suggest discussing why there was no perceptual evaluation.
